# Unified mechanisms for self-RNA recognition by RIG-I Singleton-Merten syndrome variants

**Charlotte Lässig[1,2], Katja Lammens[1,2], Jacob Lucián Gorenflos López[1,2], Sebastian Michalski[1,2], Olga Fettscher[1,2], Karl-Peter Hopfner[1,2,3]***

[1]Department of Biochemistry, Ludwig-Maximilians-Universität München, Munich, Germany; [2]Gene Center, Ludwig-Maximilians-Universität München, Munich, Germany; [3]Center for Integrated Protein Science Munich, Munich, Germany

**Abstract** The innate immune sensor retinoic acid-inducible gene I (RIG-I) detects cytosolic viral RNA and requires a conformational change caused by both ATP and RNA binding to induce an active signaling state and to trigger an immune response. Previously, we showed that ATP hydrolysis removes RIG-I from lower-affinity self-RNAs (*Lässig et al., 2015*), revealing how ATP turnover helps RIG-I distinguish viral from self-RNA and explaining why a mutation in a motif that slows down ATP hydrolysis causes the autoimmune disease Singleton-Merten syndrome (SMS). Here we show that a different, mechanistically unexplained SMS variant, C268F, which is localized in the ATP-binding P-loop, can signal independently of ATP but is still dependent on RNA. The structure of RIG-I C268F in complex with double-stranded RNA reveals that C268F helps induce a structural conformation in RIG-I that is similar to that induced by ATP. Our results uncover an unexpected mechanism to explain how a mutation in a P-loop ATPase can induce a gain-of-function ATP state in the absence of ATP.

DOI: https://doi.org/10.7554/eLife.38958.001

*For correspondence:
hopfner@genzentrum.lmu.de

**Competing interests:** The authors declare that no competing interests exist.

## Introduction

Retinoic acid-inducible gene I (RIG-I)-like receptors (RLRs) are cytosolic innate immune sensors that recognize viral double-stranded (ds)RNAs. RLRs (RIG-I, MDA5 and LGP2) are members of the so-called Superfamily II (SF2) helicases/translocases. They share a multi-domain architecture that consists of a central SF2 ATPase domain accompanied by two N-terminal tandem caspase activation and recruitment domains (2CARD, only in RIG-I and MDA5) and a C-terminal regulatory domain (CTD or RD) (*Rawling and Pyle, 2014*). The SF2 domain itself is built out of an N-terminal RecA-like domain (1A) and a C-terminal RecA-like domain (2A) that together form an ATP-binding pocket, as well as an insertion domain (domain 2B). SF2 and RD are crucial for RNA-recognition of RLRs, whereas 2CARD communicates successful RNA-binding events to downstream signaling factors (*Jiang et al., 2011*; *Kowalinski et al., 2011*; *Luo et al., 2011*). Specifically, RIG-I detects dsRNA ends harbouring 5' tri- or diphosphates (*Goubau et al., 2014*; *Schlee et al., 2009*; *Schmidt et al., 2009*). Simultaneous binding of an RNA ligand and ATP to RIG-I switches the protein into an active state in which the otherwise shielded 2CARD is released (*Zheng et al., 2015*). Activated RIG-I homo-tetramerizes via 2CARD (*Jiang et al., 2012*) and nucleates the polymerisation of its adapter protein, mitochondrial antiviral-signaling (MAVS), to elicit the innate immune signaling cascade (*Peisley et al., 2014*; *Wu et al., 2014*).

The similarity of epitopes of viral RNA recognized by RLRs, in particular dsRNA stems, to some endogenous ribonucleic acids has required the immune system to develop mechanisms besides merely recognizing 5'-di/triphosphate-containing RNA ends to discriminate self from non-self. Self-

ribonucleic acids for instance are shielded from RLRs by introducing 2'O-methylations or by destabilizing double-stranded parts through A-to-I editing (*Chung et al., 2018*; *Devarkar et al., 2016*; *Liddicoat et al., 2015*; *Schuberth-Wagner et al., 2015*). In addition, we and others have been able to show that the SF2 domain of RLRs itself confers a proof-reading activity by removing RIG-I from self-RNA so as to avoid autoimmunity (*Anchisi et al., 2015*; *Lässig et al., 2015*; *Louber et al., 2015*; *Rawling et al., 2015*). In particular, ATP turnover can lead to translocation on dsRNA stems that could remove the protein from the RNA and reinstall the inactivated state (*Myong et al., 2009*; *Yao et al., 2015*).

Deficiencies in any of these mechanisms can lead to immune recognition of self-RNAs (*Ahmad et al., 2018*; *Chiang et al., 2018*). For instance, single-nucleotide polymorphisms (SNPs) in RLR genes are known to cause system-wide autoimmune diseases such as Aicardi-Goutière syndrome (AGS) (*Oda et al., 2014*; *Rice et al., 2014*), Singleton-Merten syndrome (SMS) (*Jang et al., 2015*; *Rutsch et al., 2015*), systemic lupus erythematosus (SLE) (*Cunninghame Graham et al., 2011*; *Pettersson et al., 2017*; *Van Eyck et al., 2015*) or type 1 diabetes (*Liu et al., 2009*; *Smyth et al., 2006*). The molecular basis for the development of these diseases is in many cases not understood, because structural data on these RLR variants have been missing.

In this work, we present the biochemical and structural analysis of the RIG-I SMS variant C268F and unveil an ATP-independent signaling mechanism. We show that active site rearrangements of several amino acid side chains in RIG-I C268F mimic an ATP-bound state and activate the protein for signaling upon recognition of RNA ligands in the absence of ATP.

## Results and discussion

The RIG-I SMS variants C268F and E373A are located within SF2 ATP binding and hydrolysis motifs I and II, respectively, (*Fairman-Williams et al., 2010*) (*Figure 1—figure supplement 1*) and were previously shown to be constitutively active in reporter cells without any external dsRNA trigger (*Jang et al., 2015*; *Lässig et al., 2015*) (*Figure 1—figure supplement 2A*). RIG-I E373A's enhanced immune signaling ability can be explained by proficient ATP binding but reduced ATP hydrolysis, since the motif II glutamate is implicated in positioning and polarizing the attacking water molecule in the hydrolysis reaction. The stabilized ATP state leads to increased binding of and activation by endogenous dsRNA (*Lässig et al., 2015*; *Louber et al., 2015*). By contrast, the molecular basis for activation of RIG-I C268F is still unknown as this mutation is located in motif I, which is normally associated with ATP binding and which is critical for RIG-I activation. For instance, a designed and widely used mutant with a defect in the invariant ATP phosphate-binding motif I lysine, K270A, possesses reduced ATP binding properties (*Rawling et al., 2015*) and the RIG-I K270A/I mutant is deficient in inducing an immune response (*Lässig et al., 2015*; *Yoneyama et al., 2005*). By contrast, the RIG-I SMS variant C268F shows the opposite effect and is constitutively active, although this mutation is only two amino acids away from K270A in the same ATP phosphate-binding 'P-loop' in motif I (*Figure 1—figure supplement 2A*).

To analyze whether the autoimmune activity of the SMS variant RIG-I C268F is RNA-dependent, we designed a double-mutant that is defective in RNA-binding and carried out interferon (IFN)-β-promoter-driven luciferase reporter assays with overexpressed proteins in HEK293T RIG-I KO cells (*Figure 1A*). In particular, we used a previously described T347A point mutation in the RNA-binding interface of the N-terminal RecA-like domain (1A) (*Figure 1—figure supplement 1*), that abrogates the signaling of wild type (wt) RIG-I in infected cells and decreases the affinity for dsRNA in vitro (*Lässig et al., 2015*). Our assays show, that RIG-I C268F, T347A fails to induce the IFN-β promoter in uninfected as well as in 19mer 5'-triphosphate (ppp)-dsRNA-stimulated cells, indicating that the RIG-I SMS single-mutant C268F induces immune signaling only if bound to endogenous or transfected RNA. In addition, competition assays of RIG-I C268F titrated with signaling-deficient RIG-I lacking the 2CARD domain (RIG-I Δ2CARD) or with RIG-I Δ2CARD E373Q also gradually decreased the IFN-β promoter-driven luciferase activity (*Figure 1—figure supplement 2B*). Thus, RIG-I C268F as well as RIG-I Δ2CARD (E373Q) seem to recognize identical RNA substrates which are, however, saturated with signaling-deficient RIG-I Δ2CARD (E373Q) at higher transfected DNA concentrations thus preventing an immune response. Furthermore, we can exclude the possibility that the RIG-I

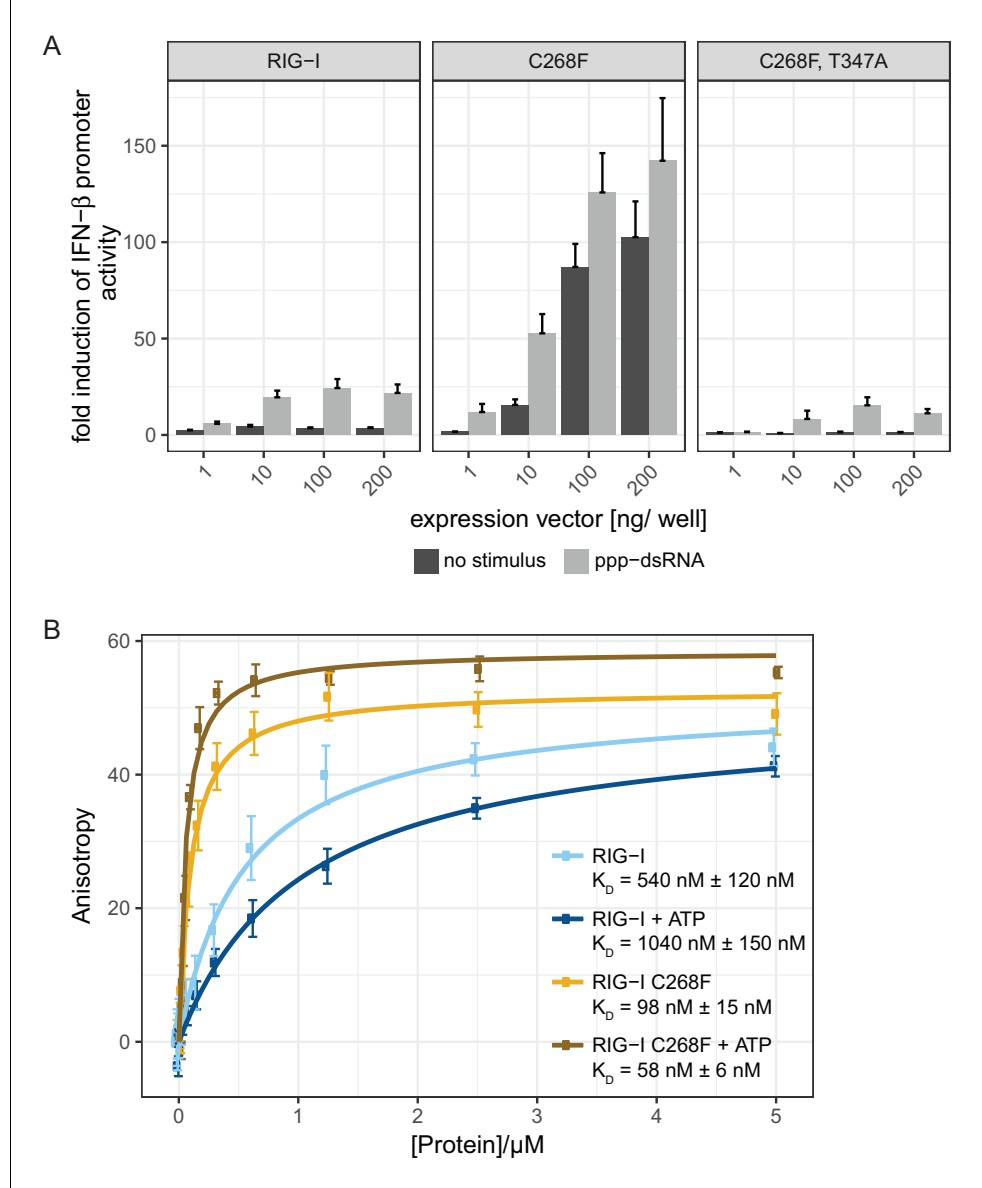

**Figure 1.** The RIG-I Singleton-Merten syndrome variant C268F signals in response to endogenous dsRNA. (**A**) Fold change of interferon (IFN)-β promoter-driven luciferase activity in uninfected HEK293T RIG-I KO cells or in cells stimulated with a 19mer 5'-triphosphate (ppp)-dsRNA upon overexpression of different RIG-I mutants. Cells were co-transfected with RIG-I expression vectors and p-125luc/pGL4.74 reporter plasmids, and stimulated with ppp-dsRNA 6 hr post transfection. Firefly luciferase activities were determined in respect to Renilla luciferase activities 16 hr after RNA stimulation. All ratios were normalized to an empty vector control. n = 4–12, error bars represent mean values + standard error of the mean (SEM). (**B**) Fluorescence anisotropy changes measured after titrating RIG-I or RIG-I C268F in the presence or absence of ATP into solutions containing a fluorescently labeled 14mer dsRNA. All binding curves were fit to a one-site binding equation using R. n = 4, error bars represent mean values ± standard deviation (SD).

DOI: https://doi.org/10.7554/eLife.38958.002

The following figure supplements are available for figure 1:

**Figure supplement 1.** Location of RIG-I amino acid substitutions used in *Figure 1*.
DOI: https://doi.org/10.7554/eLife.38958.003

**Figure supplement 2.** Comparison of the autoimmune signaling activity of RIG-I Singleton-Merten syndrome variants.
DOI: https://doi.org/10.7554/eLife.38958.004

C268F SMS mutation leads to a liberation of the 2CARD signaling module in the absence of RNA, for example by unfolding SF2. Instead, signaling by the RIG-I C268F SMS variant is dependent on 2CARD release triggered by binding to RNA molecules, as it is in wild type RIG-I.

To validate the intact RNA-binding properties of RIG-I C268F, we performed in vitro fluorescence anisotropy assays of purified proteins with a labeled hairpin (hp) RNA containing non-base-paired RNA ends (*Figure 1B*). We chose to use a double-stranded RNA ligand without blunt ends to suppress the dominant binding of RIG-I's RD to terminal RNA base pairs and to simulate recognition of endogenous-like RNA species that could be present within the cytosol. As expected, wild type RIG-I shows a moderate binding affinity to this ligand that is further decreased in the presence of ATP. By contrast, RIG-I C268F displays an already increased affinity to the hpRNA that is even enhanced in the presence of ATP. This confirms the previous results for RIG-I C268F showing an increased co-purification with endogenous RNA molecules (*Lässig et al., 2015*) and indicates that RIG-I C268F has two conformations (apo and ATP-bound) that have increased dsRNA affinity.

Since an intact SF2 ATPase domain of RIG-I is needed so that signaling only occurs when foreign RNA molecules are recognized (*Louber et al., 2015*; *Rawling et al., 2015*), we further analyzed the ATP binding and hydrolysis properties of RIG-I C268F in vitro. The ATP-binding-deficient and hydrolysis-deficient motif I mutant RIG-I K270I and the hydrolysis-deficient motif II mutant RIG-I E373Q both served as references. ATP hydrolysis assays with $^{32}$P-labeled ATP confirmed a loss of catalytic activity of RIG-I C268F and of both motif I and II mutants in the presence of dsRNA (*Figure 2A*). In accordance with our RNA-binding experiments, this evidence supports a model in which RIG-I C268F has defects in dissociating from endogenous RNA, because it lacks the capability for ATP turnover and hence translocation. To further analyze the ATP-binding properties of the RIG-I SMS variant, we conducted a tryptophan fluorescence-based FRET assay with MANT-ATP (*Rawling et al., 2015*) (*Figure 2B*). In the absence of any RNA ligand, both wild type RIG-I and ATP-hydrolysis-deficient RIG-I E373Q show comparable affinities for MANT-ATP and MANT-ATPγS in the low μM range (*Table 1*). As expected, ATP-binding-deficient RIG-I K270I has a reduced affinity for ATP (in the medium μM range). In accordance with previous data (*Kohlway et al., 2013*), the presence of RNA increases the affinity of wtRIG-I and RIG-I E373Q, but not of RIG-I K270I for MANT-ATP or MANT-ATPγS. Interestingly, like RIG-I K270I, the RIG-I SMS variant C268F displays reduced ATP-binding affinities compared to that of wtRIG-I or RIG-I E373Q independently of the availability of an RNA ligand (*Figure 2B*, *Table 1*). This is puzzling as both RNA and ATP binding are normally needed to induce a molecular switch within RIG-I in order to release the 2CARD module and to activate an immune response (*Shah et al., 2018*; *Zheng et al., 2015*). RIG-I C268F might thus be able to signal even in the absence of a bound ATP molecule. However, even though it is possible that under cellular conditions of 1 mM ATP or even higher, ATP is still bound by both motif I mutants, the molecular switch to release 2CARD is only triggered in RIG-I C268F but not in RIG-I K270I (*Figure 1—figure supplement 2A*). These peculiarities of the ATP-bound states of the motif I mutants need to be addressed in future studies.

To further investigate the influence of ATP binding to RIG-I C268F, we mutated two residues that are implicated in ATP adenine recognition, R244A and Q247A (*Figure 2—figure supplement 1*), thereby further lowering the ATP-binding affinity of the SMS mutant, and tested the ability of the resulting triple-mutant to induce an immune response (*Figure 2C*). We expected to see no decrease in the autoimmune signaling activity of the triple-mutant compared to that of the RIG-I SMS single-mutant if the single mutant was not able to bind ATP under cellular conditions. However, compared to the SMS variant itself, the IFN-β promoter activity of the ATP-binding mutant RIG-I R244A, Q247A, C268F was reduced, albeit still significantly higher than that of wild type RIG-I. Hence, these data support a model in which RIG-I C268F is able to release its 2CARD domain and start a signaling cascade by a process that is at least partly independent of ATP.

In order to elucidate the molecular basis of autoimmune signaling by RIG-I C268F, we went on to crystalize RIG-I Δ2CARD C268F in the presence of a 14mer dsRNA and ADP·BeF$_x$ or ATP and determined the corresponding structures (*Figure 3—figure supplement 1A*). Intriguingly, despite an overall high structural similarity to wtRIG-I Δ2CARD bound to dsRNA (*Jiang et al., 2011*), the SMS mutant shows crucial amino acid side chain rearrangements in the active site that stabilize the protein in an activated state but prevent binding and co-crystalization of a nucleotide (*Figure 3A*, *Figure 3—figure supplement 1B*). In particular, the bulky F268 side chain displaces the evolutionary invariant motif I K270 from its central ATP phosphate-binding position in the P-loop, into a position

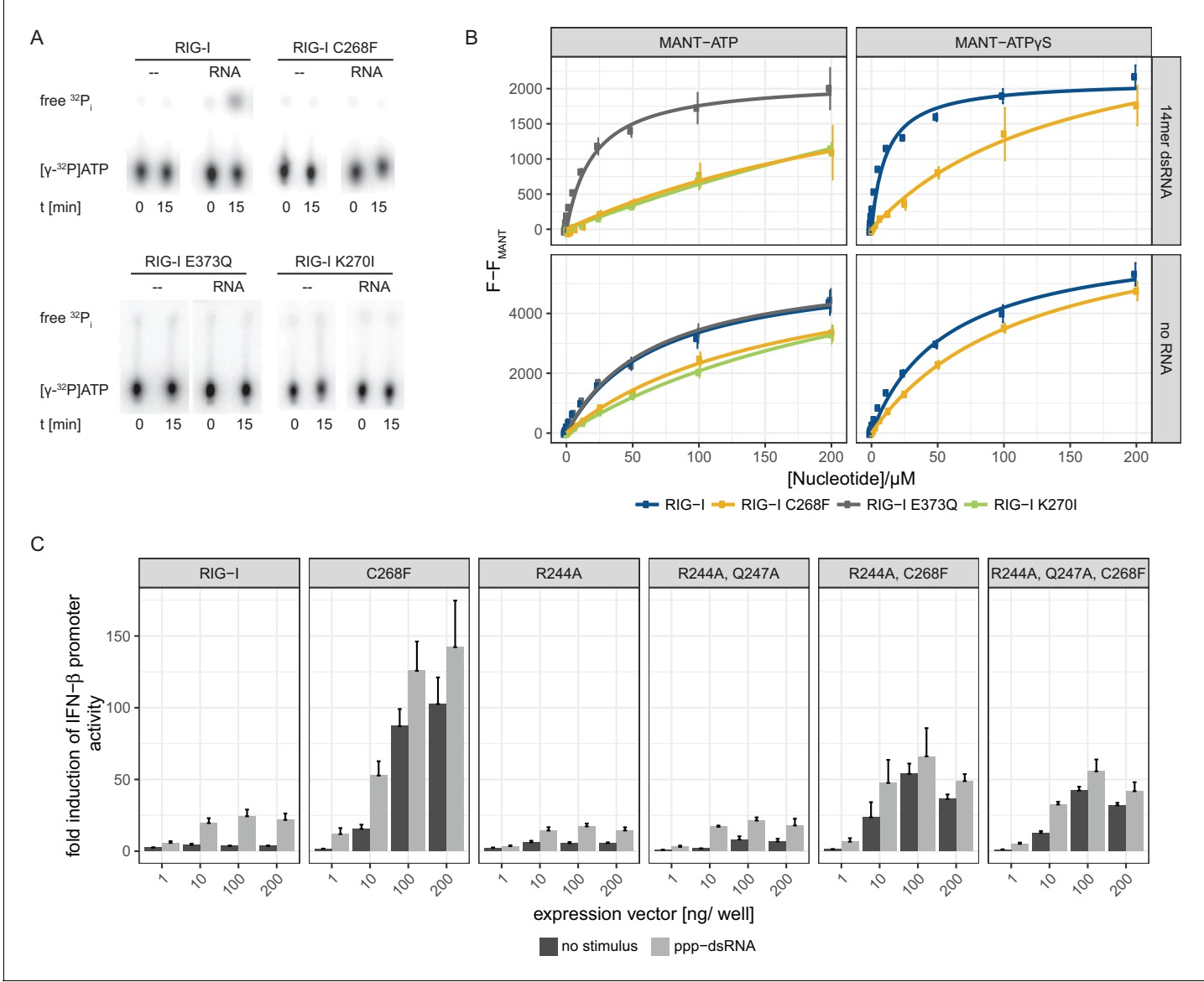

**Figure 2.** The RIG-I Singleton-Merten syndrome variant C268F is catalytically dead and has reduced ATP-binding-properties. (A) ATP hydrolysis activity of RIG-I, the RIG-I Singleton-Merten syndrome (SMS) variant C268F and the RIG-I motif I and II mutants K270I and E373Q. RIG-I proteins were incubated with [$\gamma$-$^{32}$P]-ATP in the presence or absence of a 12mer dsRNA for 15 min at room temperature and free phosphate was separated from ATP by thin layer chromatography. (B) Affinity of RIG-I, RIG-I C268F and the RIG-I motif I and II mutants to MANT-ATP or MANT-ATPγS measured by tryptophan fluorescence Förster resonance energy transfer to the MANT-nucleotide. Proteins were incubated with increasing amounts of nucleotides in the presence or absence of a 14mer dsRNA. MANT fluorescence was recorded minus a MANT-nucleotide-only control. n = 4, error bars represent mean values ± SD. (C) Fold change of interferon (IFN)-β promoter-driven luciferase activity in uninfected HEK293T RIG-I KO cells or in cells stimulated with a 19mer 5'-triphosphate (ppp)-dsRNA upon overexpression of different RIG-I mutants. Cells were co-transfected with RIG-I expression vectors and p-125luc/pGL4.74 reporter plasmids, and stimulated with ppp-dsRNA 6 hr post transfection. Firefly luciferase activities were determined in respect to Renilla luciferase activities 16 hr after RNA stimulation. All ratios were normalized to an empty vector control. n = 4–12, error bars represent mean values + SEM.

DOI: https://doi.org/10.7554/eLife.38958.005

The following figure supplement is available for figure 2:

**Figure supplement 1.** Location of RIG-I amino-acid substitutions used in *Figure 2*.

DOI: https://doi.org/10.7554/eLife.38958.006

**Table 1.** Affinities of different RIG-I mutants to MANT-ATP or MANT-ATPγS in the presence or absence of a 14mer dsRNA. n.d., not determined, n.f., no fit possible as no saturation was reached.

| Protein | MANT-ATP | MANT-ATPγS |
|---|---|---|
| RIG-I | 72 ± 13 µM | 58 ± 7 µM |
| RIG-I + RNA | n.d. | 11 ± 1 µM |
| RIG-I E373Q | 72 ± 13 µM | n.d |
| RIG-I E373Q + RNA | 28 ± 5 µM | n.d |
| RIG-I K270I | 298 ± 81 µM | n.d |
| RIG-I K270I + RNA | n.f. | n.d |
| RIG-I C268F | 166 ± 34 µM | 116 ± 13 µM |
| RIG-I C268F + RNA | n.f. | 147 ± 55 µM |

DOI: https://doi.org/10.7554/eLife.38958.007

where the Nε of K270 is situated at a site normally occupied by the Mg$^{2+}$ ion (*Video 1*). As a result, K270 now forms a salt bridge with E702 from the C-terminal RecA-like domain 2A, which in turn occupies the ATP γ-phosphate binding site. The resulting overall conformation resembles the ATP-bound state of RIG-I, but without ATP, and could thus explain how C268F is able to signal independently of any nucleotide. In order to clarify the impact of the salt bridge, we mutated both side chains in RIG-I C268F and analyzed the resulting double-mutants in our IFN-β promoter activity assay (*Figure 3B*). We expected that a disturbance of the salt-bridge formation in RIG-I C268F would lead to a loss of autoimmune signaling. Indeed, mutation of K270 in RecA-like domain A1 renders RIG-I C268F inactive, probably as the result of a mixture of (i) prevention of the activation of RIG-I C268F in the absence of ATP by disrupting the salt bridge, and/or (ii) the failure to bind ATP altogether through impaired ATP phosphate coordination and thus an impaired 2CARD release. In contrast to our ATP-binding triple mutant R244A, Q247A, C268F, which still allows formation of the salt bridge in the absence of ATP, mutation of motif I K270 in RIG-I C268F thus renders the protein inactive. Mutation of E702 in the RecA-like domain 2A of RIG-I C268F, by contrast, does not disrupt constitutive signaling of the protein, although this activity is at a reduced level compared to that of RIG-I C268F. An explanation for this might be E702's localization within the SF2 helicase motif V, which couples RNA-binding-induced ATP hydrolysis with movement on dsRNA. Similar to our previously described V699A mutant in the same motif (*Lässig et al., 2015*), E702A alone already induces an autoimmune phenotype similar to that induced by RIG-I C268F, E702A. The most plausible explanation for these data is that RIG-I C268F stabilizes an ATP-like state in the absence of ATP, but still allows formation of a proper ATP-bound state. Although E702A could reduce the former (by disrupting the salt bridge), it might stabilize the latter by sterically allowing ATP binding or by increasing the interaction with RNA.

In summary, our data suggest that the RIG-I C268F SMS mutation stabilizes the signal-on state of RIG-I in the presence of RNA but absence of ATP through a salt bridge between K270 from domain 1A and E702 from 2A. Unlike wild type RIG-I, which requires both RNA and ATP bound in order to be activated for downstream signaling, RIG-I C268F can signal independently of ATP (*Figure 2C*) in the presence of

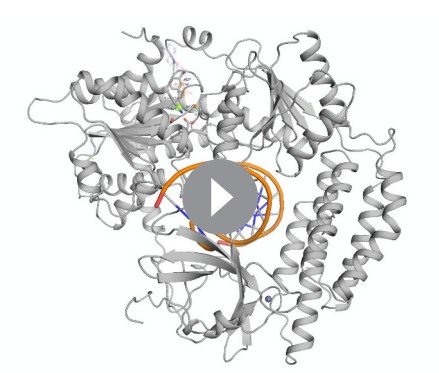

**Video 1.** Crystal structure of RIG-I Δ2CARD C268F and close-up of the active site. The Singleton-Merten syndrome (SMS) mutation F268, as well as K270 and E702, are represented by a stick model. Theoretic locations of ADP·BeF$_3$ and Mg$^{2+}$ are indicated in faint sticks and spheres, respectively, according to a superposition with RIG-I Δ2CARD in complex with RNA and nucleotide analogue (PDB 5E3H). K270 is located at the Mg$^{2+}$-binding site, whereas E702 occupies the BeF$_3$ (ATP γ-phosphate) position.
DOI: https://doi.org/10.7554/eLife.38958.010

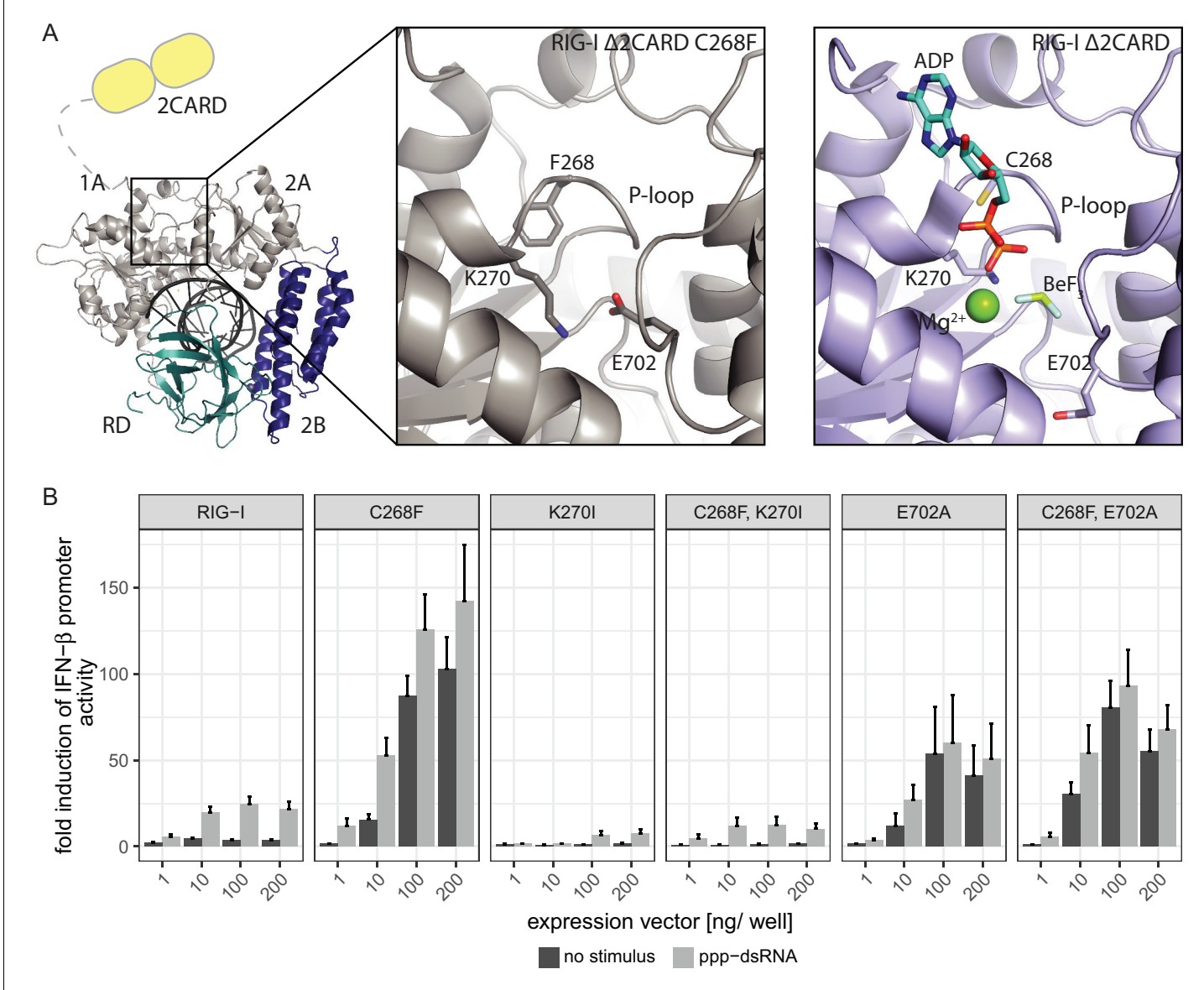

**Figure 3.** The RIG-I Singleton-Merten syndrome variant C268F induces amino acid side chain rearrangements within the active site that interfere with nucleotide binding. (A) ATP-binding pockets of the RIG-I Singleton-Merten syndrome (SMS) variant C268F (left and middle panels) and the RIG-I wild type (right panel) bound to a 14mer dsRNA. The RIG-I SF2 sub-domains are colored in light gray or light blue (1A and 2A) and dark blue (2B). The RD is depicted in cyan and 2CARD is indicated in yellow. (B) Fold change of interferon (IFN)-β promoter-driven luciferase activity in uninfected HEK293T RIG-I KO cells or in cells stimulated with a 19mer 5'-triphosphate (ppp)-dsRNA upon overexpression of different RIG-I mutants. Cells were co-transfected with RIG-I expression vectors and p-125luc/pGL4.74 reporter plasmids, and stimulated with ppp-dsRNA 6 hr post transfection. Firefly luciferase activities were determined in respect to Renilla luciferase activities 16 hr after RNA stimulation. All ratios were normalized to an empty vector control. n = 4–12, error bars represent mean values + SEM.

DOI: https://doi.org/10.7554/eLife.38958.008

The following figure supplement is available for figure 3:

**Figure supplement 1.** Structural comparison of the RIG-I Singleton-Merten syndrome variant C268F with wild type RIG-I.

DOI: https://doi.org/10.7554/eLife.38958.009

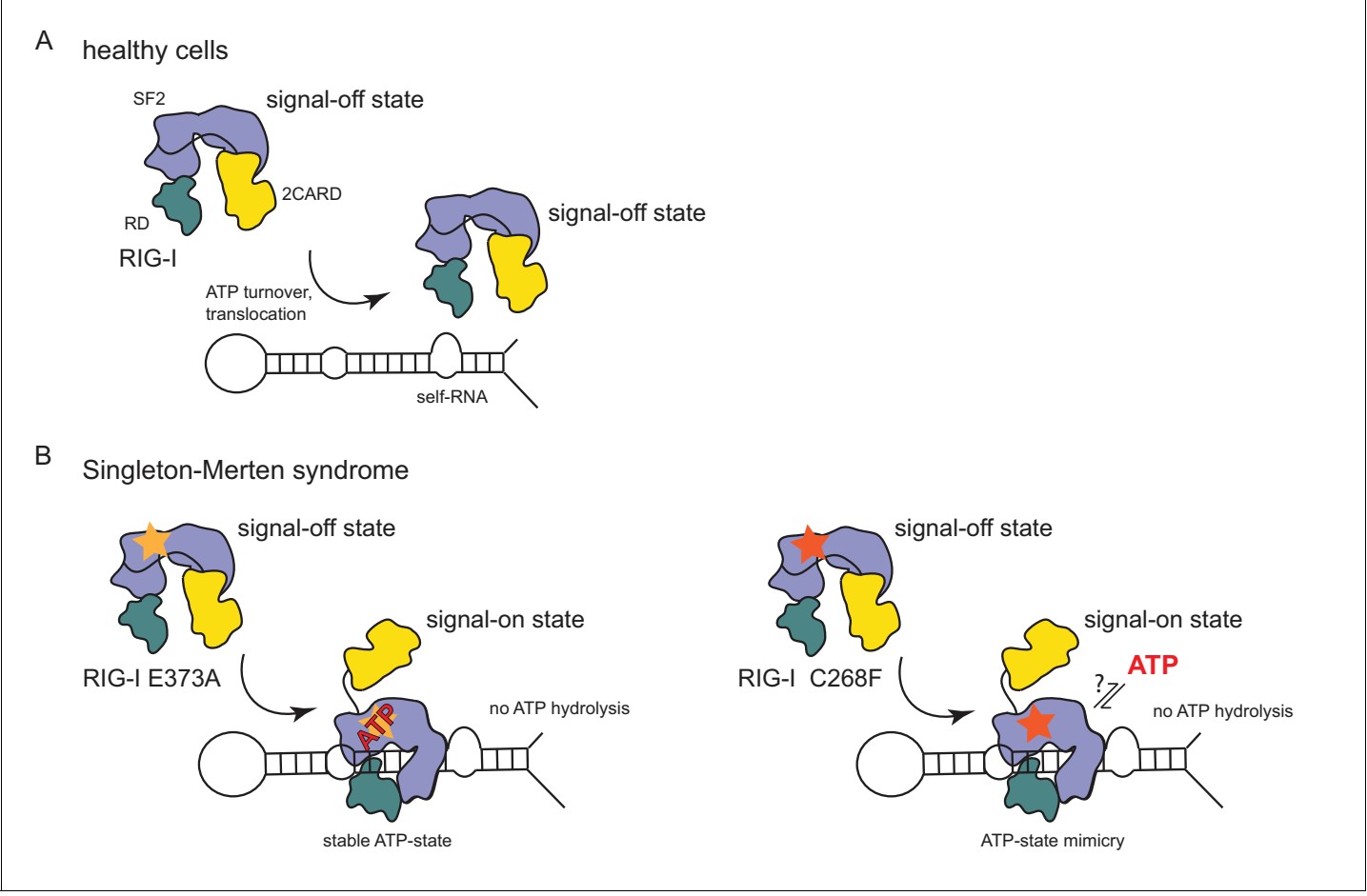

**Figure 4.** Model for the impact of Singleton-Merten syndrome mutations on self-RNA-induced RIG-I signaling. (**A**) In healthy cells, wild type RIG-I occurs in a signal-off state in which 2CARD is shielded by binding to the insertion domain of SF2. Binding of RIG-I to self-RNAs is efficiently prevented through ATP-turnover-induced dissociation (for a detailed model on self- vs non-self RNA discrimination see also *Lässig et al. (2015)*. (**B**) RIG-I Singleton-Merten syndrome (SMS) mutations either slow down ATP hydrolysis and stabilize the ATP-state (E373A, left side) or mimic the ATP-bound state (C268F, right side), and thus allow formation of the RIG-I signal-on state. In both cases, loss of ATP hydrolysis enhances the interaction with self-RNA and therefore results in pathogenic signaling. SMS mutations are indicated with a yellow or orange star.

DOI: https://doi.org/10.7554/eLife.38958.011

RNA (*Figure 1A*). However, our data also show that ATP further increases the binding of RIG-I C268F to internal dsRNA stems (*Figure 1B*) and that the engineering of ATP-binding mutations R244A, Q247A into RIG-I C268F at least partly reduces its signaling activity in response to endogenous or ppp-dsRNA RNA (*Figure 2C*). Even though we were not able to co-crystalize RIG-I C268F in the presence of a nucleotide, it is very possible that binding of ATP at high molecular concentrations, such as those present in a cellular context, further contributes to the pathogenic signaling activity of the protein. In principle, ATP binding would be sterically allowed if the salt-bridge-forming residue E702 occupies a site as in wtRIG-I. It is not yet clear what happens at the $Mg^{2+}$-binding site, as sterically F268 would not allow a canonical positioning of motif I K270 and might thus prevent binding of $Mg^{2+}$. Perhaps K270 remains at the displaced site even in the presence of ATP and simply 'mimics' $Mg^{2+}$. Such a scenario could still lead to reduced binding of ATP but would prevent ATP hydrolysis, as observed, because of a substantially altered charge distribution. This mechanism unifies the molecular basis for the development of SMS by both RIG-I variants (*Figure 4*). Unlike wtRIG-I, which consumes ATP and uses ATP turnover to decrease its affinity to self-RNA (*Figure 4A*), both motif I and II SMS mutations either mimic or freeze RIG-I in an ATP-bound state (*Figure 4B*). As a consequence, the inability to hydrolyze ATP and thus to dissociate actively from

RNA (*Figure 1B*, *Figure 2A*) traps the protein in an activated state and thus explains autoimmune signaling in response to self-RNA.

In conclusion, we provide for the first time a detailed biochemical and structural analysis of an RLR autoimmune disease variant. Slight intramolecular rearrangements within the RIG-I C268F ATP-binding pocket appear to compensate for ATP binding and render the protein active in the presence of dsRNA only. At the same time, loss of proof-reading activity leads to increased activation by endogenous RNA. This unusual gain-of-function mutation reveals, for the first time to our knowledge, that a P-loop mutation can mimic the effects of ATP binding.

# Materials and methods

**Key resources table**

| Reagent type (species) or source | Designation | Source or reference | Identifiers | Additional information |
|---|---|---|---|---|
| Cell line (human) | HEK293T RIG-I KO | *Zhu et al. (2014)* | | Growth in Dulbecco's Modified Eagle Medium (DMEM) supplemented with 10% fetal bovine serum (FBS) as monolayer |
| Strain, strain background (*Escherichia coli*) | BL21 (DE3) Rosetta | Novagene | | |
| Strain, strain background (*Escherichia coli*) | DH10multiBac | GenevaBiotech | | |
| Strain, strain background (*Spodoptera frugipeda*) | Sf21 insect cells | Thermo Fisher Scientific | 11497013 | Growth in SF-900 III serum-free medium |
| Strain, strain background (*Trichoplusia ni*) | High Five insect cells | Thermo Fisher Scientific | B85502 | Growth in Express Five serum-free medium supplemented with 10 mM L-glutamine |
| Recombinant DNA reagent | pcDNA5/FRT/TO | Thermo Fisher Scientific | V652020 | |
| Recombinant DNA reagent | pcDNA5/FRT/TO-FLAG/HA-RIG-I and various mutants of the same construct | *Lässig et al. (2015)* and this paper | | Progenitors: PCR, DDX58 (cDNA) and pcDNA5/FRT/TO |
| Recombinant DNA reagent | p-125luc | *Yoneyama et al. (1996)* | | Firefly luciferase controlled by an interferon-β promoter |
| Recombinant DNA reagent | pGL4.74 | Promega | E6921 | Constitutive expression of a Renilla luciferase |
| Recombinant DNA reagent | pFBDM | *Berger et al. (2004)* | | |
| Recombinant DNA reagent | pFBDM-His-RIG-I and various mutants of the same construct | *Lässig et al. (2015)* and this paper | | Progenitors: PCR, DDX58 (cDNA) and pFBDM |
| Recombinant DNA reagent | pETM11-SUMO3GFP | EMBL Heidelberg, H. Besir | https://www.embl.de/pepcore/pepcore_services/cloning/sumo/ | |
| Recombinant DNA reagent | pETM11-SUMO3-RIG-I-Δ2CARD-C268F | This paper | | Progenitors: PCR, DDX58 (cDNA) and pETM11-SUMO3GFP |
| Sequence-based reagent | 19mer 5' triphosphate dsRNA | InvivoGen | tlrl-3prna | 1 µg/mL, 5'-pppGCAUGC GACCUCUGUUUGA-3 |
| Sequence-based reagent | 14mer dsRNA | Dharmacon | | 5'-CGACGCUAGCGUCG-3' |
| Sequence-based reagent | Cy3-hpRNA | Biomers | | 5'-Cy3-CCACCCGCCCCCCUAGU GAGGGGGGCGGGCC-3' |
| Chemical compound, drug | Lipofectamine 2000 | Thermo Fisher Scientific | 11668019 | Used at 2.5x excess compared to RNA/DNA mass |

*Continued on next page*

*Continued*

| Reagent type (species) or source | Designation | Source or reference | Identifiers | Additional information |
|---|---|---|---|---|
| Chemical compound, drug | MANT-ATP | Jena Bioscience | NU-202 | |
| Chemical compound, drug | MANT-ATPγS | Jena Bioscience | NU-232 | |
| Chemical compound, drug | [γ-$^{32}$P]ATP | Hartmann Analytic | SRP-301 | 10 nM spiked with 3 mM unlabeled ATP |
| Commercial assay or kit | Dual-Luciferase Reporter Assay System | Promega | E1910 | |
| Software, algorithm | XDS, XSCALE | *Kabsch (2010)* | http://xds.mpimf-heidelberg.mpg.de/ | |
| Software, algorithm | PHASER | *McCoy et al. (2007)*; *Winn et al. (2011)* | http://www.ccp4.ac.uk/ | |
| Software, algorithm | Coot | *Emsley et al. (2010)* | https://www2.mrc-lmb.cam.ac.uk/personal/pemsley/coot/ | |
| Software, algorithm | PHENIX | *Afonine et al. (2012)* | https://www.phenix-online.org/ | |
| Software, algorithm | Pymol | Schrödinger | https://pymol.org/2/ | |
| Software, algorithm | R | *R Development Core Team (2013)* | https://www.r-project.org/ | |

## Vectors and cell lines

Sequences encoding full-length (1-925) or N-terminal truncated (230–925 or 232–925) human RIG-I were cloned into either pcDNA5/FRT/TO (purchased from Thermo Fisher Scientific, Waltham, MA; for expression in human cells), pFBDM (for expression in insect cells) (*Berger et al., 2004*) or pETM11-SUMO3 (EMBL, Heidelberg, Germany; for expression in *E. coli*). All proteins that were overexpressed in human cells contained an N-terminal FLAG/HA-tag, whereas proteins purified from insect cells contained an N-terminal His-tag.

Mutants were generated by site-directed mutagenesis using the QuikChange protocol and PfuUltra polymerase (Agilent, Santa Clara, CA).

HEK293T RIG-I KO cells (*Zhu et al., 2014*) were maintained in high glucose Dulbecco's Modified Eagle Medium (DMEM) supplemented with GlutaMAX, pyruvate and 10% fetal bovine serum (FBS) (all purchased from Thermo Fisher Scientific, Waltham, MA) at 37°C/ 5% $CO_2$ and were regularly tested by PCR for potential mycoplasma contaminations. *Spodoptera frugipeda* Sf21 and *Trichoplusia ni* High Five insect cells were maintained at 27°C/ 150 rpm in SF-900 III serum-free medium and High Five serum-free medium supplemented with 10 mM L-glutamine, respectively (both purchased from Thermo Fisher Scientific, Waltham, MA).

## Luciferase reporter-gene assays

Transfection-based reporter gene assays in HEK293T RIG-I KO cells were carried out in 96-well tissue culture plates seeded with $0.2 \times 10^5$ cells one day prior to transfection. Cells were transfected with 25 ng p-125Luc (inducible-expression of a Firefly luciferase controlled by an interferon-β promoter) (*Yoneyama et al., 1996*), 5 ng pGL4.74 (constitutive-expression of Renilla luciferase, Promega, Madison, WI) and varying concentrations of FLAG/HA-RIG-I plasmids (1–200 ng) in a total amount of 300 ng DNA per well (filled up with empty pcDNA5 FRT/TO) using OptiMEM and Lipofectamine 2000 (both Thermo Fisher Scientific, Waltham, MA) according to the vendor's protocol. After 6 hr, cells were either stimulated by transfection of 1 μg/mL 19mer 5'triphosphate-dsRNA (InvivoGen, San Diego, CA) in OptiMEM using Lipofectamine 2000 or the respective amount of OptiMEM alone was added. Cells were harvested 16 hr after RNA stimulation in 50 μL passive lysis buffer (Promega, Madison, WI) and frozen at −20°C until they were used. Luciferase activities were determined with a Berthold Luminometer in black 96-well plates using 20 μL cell lysate and the Dual-Luciferase Reporter Assay System (Promega, Madison, WI).

For competition assays of FLAG/HA-RIG-I C268F with N-terminally shortened FLAG/HA-RIG-I Δ2CARD (230–925), cells were seeded as described above. Cells were transfected with 25 ng p-125Luc, 5 ng pGL4.74, 75 ng FLAG/HA-RIG-I C268F plasmid and varying concentrations (1–195 ng) of competitor plasmids containing N-terminal truncated RIG-I in a total of 300 ng DNA per well (filled up with empty pcDNA5 FRT/TO) using OptiMEM and Lipofectamine 2000 as transfection reagent according to the vendor's protocol. Cells were harvested 16 hr after transfection in 50 µL passive lysis buffer and frozen at −20°C. Luciferase activities were determined as described above.

All cell-based assays were performed at least four times in independent experiments and are represented as mean values + SEM.

## Protein expression and purification

Recombinant full-length human RIG-I or RIG-I single amino acid mutants (1–925, N-terminal His-tag) were produced in and purified from High Five insect cells as described before (*Cui et al., 2008*; *Lässig et al., 2015*; *Rawling et al., 2015*). Briefly, the open reading frame for human RIG-I was cloned into the pFBDM vector, transformed into *E. coli* DH10MultiBac, and extracted Baculovirus DNA was then transfected into SF21 insect cells. Baculoviruses were propagated twice in SF21 insect cells and subsequently used for infection of High Five insect cells. Expression was carried out for 3 days at 27°C. Harvested cells were shock frozen in liquid nitrogen and stored at −20°C until they were used. For purification, cells were resuspended in lysis buffer (25 mM HEPES, 500 mM NaCl, 10 mM imidazole, 10% glycerol, 5 mM β-mercaptoethanol, pH 7) and lyzed by sonication. Cleared lysate was loaded onto Ni-NTA agarose resin (Qiagan, Hilden, Germany), washed with lysis buffer containing 300 mM NaCl and eluted in elution buffer (25 mM HEPES, 100 mM NaCl, 200 mM imidazole, 10% glycerol, 5 mM β-mercaptoethanol, pH 7). Proteins were further purified on a HiTrap Heparin HP column (GE Healthcare, Little Chalfont, UK) in 25 mM HEPES, 10% glycerol, 5 mM β-mercaptoethanol, pH 7 using a linear salt gradient ranging from 100 mM to 1 M NaCl. Finally, fractions containing RIG-I were pooled and loaded onto a HiLoad Superdex 200 16/60 size exclusion column (GE Healthcare, Little Chalfont, UK) using gel filtration buffer (25 mM HEPES, 150 mM NaCl, 5 mM MgCl$_2$, 5% glycerol, 5 mM β-mercaptoethanol, pH 7). Monomeric RIG-I was concentrated to ~6 mg/mL, flash frozen in liquid nitrogen and stored at −80°C. N-terminally truncated RIG-I Δ2CARD C268F (232–925, N-terminal His-tag) expressed in High Five cells was purified as described above.

In addition, N-terminally truncated RIG-I Δ2CARD C268F (232–925, N-terminal His-Sumo3-tag) was produced in and purified from *E. coli* Rosetta (DE3). Cells were induced with 0.2 mM DTT at an OD$_{600}$ of 0.6–0.8 and protein was expressed at 18°C overnight. Harvested cells were shock frozen in liquid nitrogen and stored at −20°C. Protein was purified as described above, except that after metal affinity chromatography, the His-SUMO-tag was cleaved off by adding SenP2 protease (mass ratio 1:500) to pooled eluate fractions during a dialysis step against elution buffer without imidazole (25 mM HEPES, 100 mM NaCl, 10% glycerol, 5 mM β-mercaptoethanol, pH 7) that was carried out overnight at 4°C. Cleaved protein was separated from the tag during a second metal-affinity chromatography by step-wise elution with elution buffers containing 20 mM and 40 mM imidazole, and subjected to heparin-affinity and gel-filtration chromatography as described above.

For crystallization, N-terminally truncated RIG-I Δ2CARD C268F (purified either from insect cells or from *E. coli*) was concentrated to ~25 mg/mL.

## Protein crystalization

Co-crystalization of RIG-I Δ2CARD C268F (170 µM) with equimolar concentrations of 14mer dsRNA (5'-CGACGCUAGCGUCG-3', palindromic RNA, purchased from Dharmacon, Lafayette, CO) was done either in the presence of 2 mM ADP, 2 mM BeCl$_2$ and 10 mM NaF to reconstitute ADP·BeF$_x$ (crystal 1) or with 2 mM ATP (crystal 2) by hanging-drop vapor diffusion at 20°C. In each case, 2.5 µL protein/RNA/nucleotide mix was added to 2.5 µL reservoir solution from a total reservoir volume of 400 µL per well. Crystal 1 was raised in wells containing 0.1 M MOPS pH 7.5, 15% (w/v) PEG 3350, 0.125 M NaSCN and 3% (v/v) 2,2,2-trifluoroethanol as reservoir solution. The reservoir of crystal 2 contained 0.1 M MOPS pH 7.5, 17.5% (w/v) PEG 3350, 0.25 M NaSCN and 3% (v/v) 2,2,2-trifluoroethanol. Crystals appeared after 1–2 days and were transferred into the respective reservoir solutions containing 10% (v/v) 2,3-butanediol as cryoprotectant, flash-frozen and stored in liquid nitrogen.

## Data collection and structure determination

X-ray diffraction data were collected at the SLS X06SA beamline (Swiss Light Source, Villigen, Switzerland). Diffraction datasets from both crystals were indexed and integrated using XDS and scaled with XSCALE (*Kabsch, 2010*). Crystal 1 had space group $P2_12_12_1$ and diffracted to 3.3 Å, crystal 2 had space group $P6_522$ and diffracted until 2.9 Å (*Table 2*). Diffraction data from crystal 1 were used to determine an initial structure of RIG-I Δ2CARD C268F by molecular replacement using PHASER (*McCoy et al., 2007*; *Winn et al., 2011*) and a search model based on a published structure of RIG-I Δ2CARD (PDB entry 5E3H) (*Jiang et al., 2011*). The initial model was created in two iterative rounds of manual model building and refinement using Coot and PHENIX (*Afonine et al., 2012*; *Emsley et al., 2010*). This model was used to phase the second, better-diffracting dataset from crystal 2 using PHASER. The final structure was built and refined in several iterative rounds using Coot and PHENIX. The statistics describing both structures are shown in (*Table 2*). We did not detect any density for a bound nucleotide in either structure.

**Table 2.** Data collection and refinement statistics.
Values in parentheses are for the highest resolution shell.

|  | Crystal 1 | Crystal 2 |
|---|---|---|
| PDB code |  | 6GPG |
| **Data collection** |  |  |
| Space group | $P2_12_12_1$ | $P6_522$ |
| Wavelength (Å) | 1.00 | 1.00 |
| Cell dimensions |  |  |
| $a$, $b$, $c$ (Å) | 112.1, 177.1, 314.8 | 175.6, 175.6, 109.5 |
| $\alpha$, $\beta$, $\gamma$ (°) | 90, 90, 90 | 90, 90, 120 |
| Resolution range (Å) | 47.2–3.3 (3.42–3.30) | 46.4–2.9 (3.00–2.89) |
| $R_{merge}$ (%) | 14.3 (112) | 7.6 (206) |
| $I/\sigma I$ | 8.45 (1.28) | 19.72 (1.15) |
| $CC_{1/2}$ | 99.8 (67.6) | 99.9 (99.7) |
| Completeness (%) | 95.3 (79.7) | 99.7 (97.4) |
| Redundancy | 3.38 (2.91) | 13.09 (13.21) |
| **Refinement** |  |  |
| Resolution (Å) | 3.3 | 2.9 |
| No. reflections | 90,121 | 22,649 |
| $R_{work}$/ $R_{free}$ | 22.8/28.3 | 21.4/25.9 |
| No. atoms |  |  |
| Macromolecules | 35,730 | 5,810 |
| Ions | 10 | 2 |
| Ramachandran statistics |  |  |
| Favoured (%) | 92.78 | 92.71 |
| Allowed (%) | 6.39 | 6.98 |
| Outliers (%) | 0.83 | 0.31 |
| R.M.S deviations |  |  |
| Bond lengths (Å) | 0.011 | 0.009 |
| Angles (°) | 1.48 | 1.43 |
| B-factors |  |  |
| Macromolecules | 109.98 | 139.89 |
| Ions | 105.23 | 121.74 |

DOI: https://doi.org/10.7554/eLife.38958.012

Figures and movies were created with PyMOL (*Schrödinger, 2015*).

## ATP hydrolysis assays

ATP hydrolysis activities of different full-length His-RIG-I constructs were determined using [$\gamma$-$^{32}$P] ATP (Hartmann Analytic, Braunschweig, Germany). 100 nM protein was pre-incubated with 100 nM 12mer 5'triphosphate-dsRNA for 10 min at room temperature in hydrolysis buffer (25 mM HEPES, 50 mM KCl, 5 mM MgCl$_2$, 5 mM TCEP, pH 7.5). The reaction was initiated by the addition of 3 mM unlabeled and 10 nM [$\gamma$-$^{32}$P]ATP and incubated for 15 min at 37°C. Free phosphate was separated from ATP by thin layer chromatography (TLC) in TLC running buffer (1 M formic acid, 0.5 M LiCl) on polyethyleneimine cellulose TLC plates (Sigma-Aldrich, St. Louis, MO). [$\gamma$-$^{32}$P]P$_i$ and [$\gamma$-$^{32}$P]ATP were detected using the Typhoon$^{TM}$ FLA 9500 phosphor-imaging system (GE Healthcare, Little Chalfont, UK).

## Fluorescence anisotropy

Affinities of full-length RIG-I or RIG-I C268F (1–925, N-terminal His-tag) to a Cy3-labeled hpRNA (5'-Cy3-CCACCCGCCCCCCUAGUGAGGGGGGCGGGCC-3', purchased from Biomers, Ulm, Germany) were determined using fluorescence anisotropy. All samples were prepared in 96-well black chimney microplates (Greiner Bio-One, Kremsmünster, Austria). 10 nM RNA was pre-incubated with different protein concentrations (2.4 nM – 5 µM) for 15 min at room temperature in assay buffer (25 mM HEPES, 50 mM KCl, 5 mM MgCl$_2$, 1 mM DTT, pH 7). Fluorescence anisotropy was measured after the addition of 5 mM ATP using a TECAN M1000 microplate reader (Tecan, Männedorf, Switzerland) at $\lambda_{ex}/_{em}$ = 530/570 nm and gain = 130 during a time course of 25 min in 1 min intervals. Assays were performed four times in independent experiments using the same protein purification batch. For determination of affinities, anisotropy values between 15 and 20 min measuring time were averaged and fit to the following single-site binding model using R (*R Development Core Team, 2013*):

$$y = B_{max} \frac{[P]}{[P] + K_D} \tag{1}$$

where y is the observed anisotropy, [P] is the protein concentration, B$_{max}$ is the maximal anisotropy and K$_D$ is the dissociation constant. Fitting was performed globally on all available datasets. Representative values in figures are mean values ± SD.

## MANT-ATP binding

Binding of MANT-ATP and MANT-ATPγS to different full-length RIG-I mutants (1–925, N-terminal His-tag) was determined via Förster resonance energy transfer from RIG-I to MANT-ATP (Jena Bioscience, Jena, Germany). All samples were prepared in 96-well black chimney microplates. 2.2 µM protein and equimolar concentrations of a 14mer dsRNA (5'-CGACGCUAGCGUCG-3', see 'Protein crystalization') were pre-incubated with different MANT-ATP concentrations (0.2 µM – 200 µM) for 15 min at room temperature in assay buffer (25 mM HEPES, 50 mM NaCl, 5 mM MgCl$_2$, 1 mM DTT, pH 7). Fluorescence of MANT-ATP was measured in a TECAN M1000 microplate reader at $\lambda_{ex}/_{em}$ = 290/448 nm, gain = 170 using an average of five reads per well. Assays were performed four times in independent experiments using the same protein purification batch. For determination of affinities fluorescence values were fitted to *Equation (1)* using R, where y is the fluorescence, [P] is the protein concentration, B$_{max}$ is the maximal fluorescence and K$_D$ is the dissociation constant. Fitting was performed globally on all available datasets. Representative values in figures are mean values ± SD.

## Acknowledgements

We thank the Swiss Light Source (Villigen) Beamline scientists for their excellent technical assistance. We also thank M Moldt for help with insect cells, V Hornung for the HEK293T RIG-I KO cell line, T Fujita for the p-125luc plasmid, F Civril for the RIG-I insect cell expression vector, F Schlauderer and A Alt for help with crystalization and the whole Hopfner group for discussions.

## Additional information

### Funding

| Funder | Grant reference number | Author |
|---|---|---|
| Bayerisches Staatsministerium für Bildung und Kultus, Wissenschaft und Kunst | BioSysNet | Karl-Peter Hopfner |
| Deutsche Forschungsgemeinschaft | CIPSM | Karl-Peter Hopfner |
| Deutsche Forschungsgemeinschaft | HO2489/8 | Karl-Peter Hopfner |
| Deutsche Forschungsgemeinschaft | CRC1054 project B02 | Katja Lammens |
| Deutsche Forschungsgemeinschaft | CRC/TRR 237 | Karl-Peter Hopfner |

The funders had no role in study design, data collection and interpretation, or the decision to submit the work for publication.

### Author contributions

Charlotte Lässig, Conceptualization, Formal analysis, Supervision, Validation, Investigation, Visualization, Methodology, Writing—original draft, Writing—review and editing; Katja Lammens, Formal analysis, Supervision, Investigation, Methodology; Jacob Lucián Gorenflos López, Sebastian Michalski, Investigation, Writing—review and editing; Olga Fettscher, Investigation, Cloned and tested mutant proteins for cell-based assays; Karl-Peter Hopfner, Conceptualization, Supervision, Funding acquisition, Writing—original draft, Project administration, Writing—review and editing

### Author ORCIDs

Charlotte Lässig http://orcid.org/0000-0001-6253-7880
Katja Lammens https://orcid.org/0000-0002-4438-1381
Karl-Peter Hopfner http://orcid.org/0000-0002-4528-8357

### Decision letter and Author response

Decision letter https://doi.org/10.7554/eLife.38958.020
Author response https://doi.org/10.7554/eLife.38958.021

## Additional files

### Supplementary files

• Transparent reporting form
DOI: https://doi.org/10.7554/eLife.38958.013

### Data availability

Diffraction data have been deposited in PDB under the accession code 6GPG.

The following dataset was generated:

| Author(s) | Year | Dataset title | Dataset URL | Database, license, and accessibility information |
|---|---|---|---|---|
| Lässig C, Lammens K, Hopfner KP | 2018 | Structure of the RIG-I Singleton-Merten syndrome variant C268F | https://www.ebi.ac.uk/pdbe/entry/pdb/6gpg | Publicly available at the RCSB Protein Data Bank (accession no. 6GPG) |

The following previously published dataset was used:

| Author(s) | Year | Dataset title | Dataset URL | Database, license, and accessibility information |
|---|---|---|---|---|
| Jiang F, Miller MT, Marcotrigiano J | 2015 | Structural Basis for RNA Recognition and Activation of RIG-I | https://www.rcsb.org/structure/5E3H | Publicly available at the RCSB Protein Data Bank (accession no. 5E3H) |

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
