## [Decision Letter]

Thank you for submitting your article "Unified mechanisms for self-RNA recognition by RIG-I Singleton-Merten syndrome variants" for consideration by *eLife*. Your article has been reviewed by two peer reviewers, and the evaluation has been overseen by a Reviewing Editor and John Kuriyan as the Senior Editor. The following individual involved in review of your submission has agreed to reveal her identity: Sua Myong (Reviewer #1).

This is an interesting novel extension of their prior publication in *eLife*, structurally rationalizing the behavior of a disease-causing mutation in RIG-I.

This manuscript is acceptable without further review, but some textual clarification is requested. The authors should consider whether to do this, and if they feel that these changes would improve their manuscript they should make the changes. The revised manuscript will be accepted with a minimum of editorial intervention, without further reviewer input.

1) The manuscript is written in a condensed manner i.e., there are many mutants and various assays presented in the manuscript. It will be great if the authors can take time to explain why each mutant was made and what the expected result vs. experimental outcome was.

2) Despite the structural data in support of this idea, the ATP-independent mode of activity is less certain. It is very difficult to prove that ATP is not binding under cellular conditions. C268F is likely bound to ATP in the cell due to the high concentration of ATP (1 mM or higher), even in consideration of the binding data in Figure 2B. Binding of C268F to ATP is weak, but clearly observable. Some qualification should be noted.

*Reviewer #1:*

This is an exciting study presented by the Hopfner lab. RIG-I is a major antiviral innate immune sensor which distinguishes self- from viral RNA by recognizing the 5'-triphosphate moiety, a unique molecular signature of viral RNA. Moreover, the ATP dependent translocation activity and the subsequent release of the protein is important for discriminating the self-RNA. Understanding the mechanism by which RIG-I acts to transmit innate immune signaling is critical, especially in light of severe mutations that occur in Singleton Merton Syndrome. In this study, the authors performed a series of thorough biochemical analysis of one of the poorly understood SMS variants, C268F which causes constitutive immune activation. They explain the biochemical defect of C268F by structural analysis of the protein. The combined analysis points toward a clear structural basis of the mutant phenotype. Impressively, the authors compared three different mutants including previously characterized E373A and K270I to the current mutant, C268F. A clever approach of creating double mutant C268F/T347A demonstrated that RNA binding is still required for IFN-β induction i.e. immune activation by C268F. Both the apo and ATP-bound form of C268F mutants exhibited increased RNA binding affinity although the mutant had reduced affinity for ATP and deficient at ATP hydrolysis. Additional mutations, R244A and Q247A which further reduced ATP binding affinity still induced comparable level of immune signaling. Taken together, the C268F mutant is capable of constitutive antiviral signaling when bound to any RNA, even without ATP binding or ATP hydrolysis. The structure shows why. The C268F results in the formation of a special salt bridge in P-loop which leads to a conformation that mimics ATP bound state even in the absence of ATP. When the salt bridge is abolished, the constitutive effect disappears, signifying that the structural rearrangement induced by C268F is the key in creating an ATP bound conformation which induce the antiviral immune-signaling. The study is meticulously carried out, each hypotheses thoroughly tested and the structural result perfectly complements the biochemical data. The only point to be addressed is the following assays presented in the manuscript. It will be great if the authors can take time to explain why each mutant was made and what the expected result vs. experimental outcome was. I highly recommend publication of this work in *eLife*.

*Reviewer #2:*

Previous work from some of these authors, published in *eLife*, indicated that RIG-I utilizes its ATP hydrolysis driven translocation on double-stranded RNA to distinguish self RNA from non-self RNA. The ATPase activity allowed the enzyme to rapidly translocate from self RNA while non-self dsRNA tended to slow the ATP hydrolysis thereby allowing more stable RNA binding and subsequent signaling of the innate immune pathway. In the current submission, the authors provide a possible structural mechanism for how a known mutation in RIG-I causes the Singleton-Merten autoimmune syndrome. Biochemical data indicate little or no ATP hydrolysis and low ATP binding of the C268F variant. Structural data indicates that C268F leads to arrangement of amino acids in the ATP binding site as if ATP is bound (but in the absence of ATP). This arrangement of amino acids and lack of ATP hydrolysis leads to continued immune signaling, or in other words, a "gain-of-function". The current work therefore builds upon the previous work by showing how a mutation can lead to immune activation by causing the enzyme to mimic a structural conformation that resembles the ATP-bound or "activated" conformation. Thus, C268F RIG-I is unable to translocate off of an RNA substrate (supported by binding data). The current data does appear to solidify the proposed mechanism involving the ATP binding and hydrolysis which is a significant advance.

Summary of Concerns:

An additional significant advance claimed by the authors is that the C268F mutation can lead to RIG-I being locked into an "signal-on state" even in the absence of ATP binding. Despite the structural data in support of this idea, the ATP-independent mode of activity is less certain. It is very difficult to prove that ATP is not binding under cellular conditions. C268F is likely bound to ATP in the cell due to the high concentration of ATP (1 mM or higher), even in consideration of the binding data in Figure 2B. Binding of C268F to ATP is weak, but clearly observable. The results with the double mutant C268F, K270I seem to support the idea that ATP binding is needed for activation of the immune response, because this double mutant displays very low immune activity and likely does not bind to ATP.

The concerns are balanced by high quality structural and solid biophysical work coupled with a gene expression reporter that supports the biophysical data. The RIG-I enzyme does appear to distinguish self RNA from non-self RNA through the activation or lack of activation of the ATP hydrolysis activity or through structural conformations that mimic these conformations. I am less sure of whether this occurs in complete absence of ATP binding.

---

## [Author Response]

This manuscript is acceptable without further review, but some textual clarification is requested. The authors should consider whether to do this, and if they feel that these changes would improve their manuscript they should make the changes. The revised manuscript will be accepted with a minimum of editorial intervention, without further reviewer input.1) The manuscript is written in a condensed manner i.e., there are many mutants and various assays presented in the manuscript. It will be great if the authors can take time to explain why each mutant was made and what the expected result vs. experimental outcome was.

We very much thank Sua Myong for her positive feedback on our manuscript and for this advice. We wrote the manuscript as condensed as possible in order to satisfy the 1500 word limit stated in the guidelines. Following the advice, we added further explanations detailing the rationale behind our experimental set-up and the interpretation of the outcome. Furthermore, we explained our mutants better in the supplemental figures Figure 1—figure supplement 1 and Figure 2—figure supplement 1. Here we show the location of all used mutations in more detail.

2) Despite the structural data in support of this idea, the ATP-independent mode of activity is less certain. It is very difficult to prove that ATP is not binding under cellular conditions. C268F is likely bound to ATP in the cell due to the high concentration of ATP (1 mM or higher), even in consideration of the binding data in Figure 2B. Binding of C268F to ATP is weak, but clearly observable. Some qualification should be noted.

We also like to thank reviewer 2 for the valuable feedback and the argument concerning cellular ATP concentrations. We added a sentence mentioning cellular ATP concentrations within the passage describing the FRET ATP-binding assay (Results and Discussion, fourth paragraph) as well as within the discussion of our results (Results and Discussion, seventh paragraph).